# Prevalence of Emotional Eating in Groups of Students with Varied Diets and Physical Activity in Poland

**DOI:** 10.3390/nu14163289

**Published:** 2022-08-11

**Authors:** Mateusz Grajek, Karolina Krupa-Kotara, Agnieszka Białek-Dratwa, Wiktoria Staśkiewicz, Mateusz Rozmiarek, Ewa Misterska, Krzysztof Sas-Nowosielski

**Affiliations:** 1Department of Public Health, Faculty of Health Sciences in Bytom, Medical University of Silesia in Katowice, 41902 Bytom, Poland; 2Department of Humanistic Foundations of Physical Culture, Faculty of Physical Education, Jerzy Kukuczka Academy of Physical Education in Katowice, 40065 Katowice, Poland; 3Department of Epidemiology, Faculty of Health Sciences in Bytom, Medical University of Silesia in Katowice, 41902 Bytom, Poland; 4Department of Human Nutrition, Faculty of Health Sciences in Bytom, Medical University of Silesia in Katowice, 41902 Bytom, Poland; 5Department of Technology and Food Quality Evaluation, Faculty of Health Sciences in Bytom, Medical University of Silesia in Katowice, 41902 Bytom, Poland; 6Department of Sports Tourism, Faculty of Physical Culture Sciences, Poznan University of Physical Education, 61871 Poznan, Poland; 7Department of Pedagogy and Psychology, Faculty of Social Studies in Poznan, Poznan School of Security, 60778 Poznan, Poland

**Keywords:** emotional eating (EE), diet, physical activity, field of study, stress

## Abstract

Background: Emotional eating (EE) is not a separate eating disorder, but rather a type of behavior within a group of various eating behaviors that are influenced by habits, stress, emotions, and individual attitudes toward eating. The relationship between eating and emotions can be considered on two parallel levels: psychological and physiological. In the case of the psychological response, stress generates a variety of bodily responses relating to coping with stress. Objective: Therefore, the main objective of this study was to evaluate and compare the prevalence of emotional eating in groups of students in health-related and non-health-related fields in terms of their differential health behaviors—diet and physical activity levels. Material and Methods: The cross-sectional survey study included 300 individuals representing two groups of students distinguished by their fields of study—one group was in health-related fields (HRF) and the other was in non-health-related fields (NRF). The study used standardized questionnaires: the PSS-10 and TFEQ-13. Results: The gender of the subjects was as follows: women, 60.0% (174 subjects) (HRF: 47.1%, n= 82; NRF: 52.9%, n = 92); men, 40.0% (116 subjects) (HRF: 53.4%, n = 62; NRF: 46.6%, n = 54). The age of the subjects was 26 years (±2 years). Based on the results of the TFEQ-13, among 120 subjects (41.4%) there were behaviors consistent with limiting food intake (HRF: 72.4%; NRF: 11.0%), while 64 subjects (20.7%) were characterized by a lack of control over food intake (HRF: 13.8%, 20 subjects; NRF: 27.4%, 20 subjects). Emotional eating was characteristic of 106 students (37.9%), with the NRF group dominating (61.6%, n = 90). It was observed that a high PSS-10 score is mainly characteristic of individuals who exhibit EE. Conclusions: The results obtained in the study indicate that lifestyle can have a real impact on the development of emotional eating problems. Individuals who are characterized by elevated BMI values, unhealthy diets, low rates of physical activity, who underestimate meal size in terms of weight and calories, and have high-stress feelings are more likely to develop emotional eating. These results also indicate that further research in this area should be undertaken to indicate whether the relationships shown can be generalized.

## 1. Introduction

Emotional eating (EE) is not a separate eating disorder, but rather a type of behavior within a group of various eating behaviors that are influenced by habits, stress, emotions, and individual attitudes toward eating [1]. The relationship between eating and emotions can be considered on two parallel levels: psychological and physiological. In the case of the psychological response, stress generates a variety of bodily responses relating to coping with stress; a person under stress seeks to minimize feelings of tension accompanying given situations [2]. In physiological terms, stress, due to activation of the nervous system, causes an increase or decrease in appetite which is the basis for changes in eating behavior [1].

In a stressful situation, a person implements a series of specific activities known as stress coping. These mechanisms are aimed at changing the situation in which the individual finds himself and improving the persistent emotional state [3,4]. These actions are focused on a task-oriented approach to the problem causing stress, its solution, emotional self-regulation regarding tension, and alleviation of negative emotional states [2].

In stressful situations, eating seems to be one of the most common, simplest, and least conscious actions, and it is independent of the body weight or eating behavior of those responding in this way. In a stressful situation, eating becomes a factor in relieving emotional tension. It is used for this purpose for several reasons [3,4,5]. (1) Food is readily available nowadays, associated with a large number of grocery stores, restaurants, bars, cafes, pastry stores, and outlets where fast-food dishes and sweet and salty snacks (which persons reach for most often in a stressful situation) can be easily obtained. (2) Eating does not require the participation of other persons. The preparation of a full meal as well as a quick snack (candy, chips, crackers) does not have to depend on the presence or skill of other persons, nor on the skill of the person reducing emotional tension with food. (3) Eating is socially acceptable, which means that eating under stress does not elicit negative judgments or comments from others, unlike alcohol, cigarettes or psychoactive substances, the use of which can also stem from a desire to reduce stress. (4) Food has a strong positive connotation, mainly through associations dating back to early childhood; food is associated with the presence of the mother, a sense of security, emotional closeness, and joy.

Emotional eating (EE), unlike specific eating disorders, is not associated with a complete loss of control over the quantity and quality of food consumed. Affected individuals can stop eating at any time while experiencing the relief associated with relieving emotional tension and stress [6]. Unfortunately, because EE is not explicitly recognized as an eating disorder, but rather an eating phenomenon, there are no homogeneous diagnostic criteria, and presumptions about the prevalence of EE are based on psychometric tools popularly used in research [4]. Epidemiological data on stress eating syndrome is unknown, due to the possibly high profile of the problem, but it is known that stress eating is more common in persons with obesity [5]. It is also possible that sociodemographic and psychosocial factors such as gender, age, education, occupation, income level, stress resistance, and emotion regulation strategies have a real impact on the incidence of this condition [3]. The main exposure group, in this case, seems to be young persons who are affected by the modern rush of life and maybe more strongly exposed to stressors due to their work and education [4]. One way to counteract obesity is to expend energy through regular physical activity. In addition to the benefits of weight reduction, those who are physically active may see a reduction in low back and joint pain, improved fitness and performance, as well as improved well-being and increased self-esteem [7]. It is worth noting that physical activity plays an important role in obesity prevention not only among the elderly [8], but above all has a huge impact on shaping individuals already in childhood and adolescence, thus contributing to a reduced risk of obesity in adulthood [9]. Unfortunately, adults, due to their desire for rapid improvements in their health, often engage in risky behavior in terms of physical activity, led by the use and abuse of sports supplements [10] or the practice of unhealthy or even life-threatening diets [11]. Therefore, activities aimed at promoting physical activity among the public in a sustainable manner, for example by local governments [12] or healthcare professionals [13], are extremely important.

The main objective of this study was to evaluate and compare the prevalence of emotional eating in groups students in health related and non-health related fields in terms of their differential health behaviors—diet and physical activity levels.

The following research hypotheses were posed in preparation for the study:Emotional eating is more common among persons who have a non-rational diet.Emotional eating is more common among individuals who represent a low level of physical activity.Emotional eating is more common among persons who underestimate the size and calorie portions of foods.Emotional eating is more common among persons who exhibit high levels of daily life stress.

## 2. Materials and Methods

### 2.1. Study Background

The study is a continuation of the research presented in the paper: Grajek, M.; Krupa-Kotara, K.; Sas-Nowosielski, K.; Misterska, E.; Kobza, J. Prevalence of Orthorexia in Groups of Students with Varied Diets and Physical Activity (Silesia, Poland). Nutrients 2022, 14, 2816. https://doi.org/10.3390/nu1414281. Hence, the methodological description of the study, the characteristics of the group, and the description of the main indicators (diet, level of physical activity, ability to estimate portion size, and calorie content of a meal) are the same for both studies.

### 2.2. Sample Group

The study included 300 individuals representing equally sized groups of students from two fields of study, health-related fields (HRF) and non-health-related fields (NRF). The sample size was estimated based on the minimum sample size formula, and the data substituted from the formula took into account the total number of students of a given year at a given university. This ensured that a representative group of survey participants was achieved. The survey questionnaire was directed to all students of a particular year and field of study. The return rate of the questionnaire was estimated at 82.5%.

All subjects were students in the final year of their master’s degree (second year of their sophomore year):HRF group (144 subjects): dietetics (Medical University of Silesia in Katowice) was studied by 48.6% of the subjects (n = 80), and physical education (Academy of Physical Education in Katowice) by 51.4% of the subjects (n = 74).NRF group (146 subjects): management (University of Economics in Katowice) was studied by 47.3% of the subjects (n = 69), and computer science (Silesian University of Technology) by 52.7% (n = 77).

Based on an abbreviated medical history, it was noted that 5.2% (15 subjects) were diagnosed with chronic diseases; these were seasonal allergies—diseases that do not significantly affect their lifestyles. The main addiction in the surveyed groups was smoking, to which 3.8% of students (11 persons) admitted. No persons compulsively consumed alcohol or took other psychoactive drugs.

### 2.3. Eligibility Criteria

The HRF group consisted of 150 final-year students with majors in dietetics and physical education. The rationale for selecting this group was the fact that they have in-depth and professional knowledge in the field of rational nutrition and physical activity. The NRF group consisted of 150 students in their final year of second-degree studies with majors in management and computer science. The rationale for selecting this group was the fact that they did not have in-depth and professional knowledge in the field of rational nutrition and physical activity, at least at the university level. The assumption for the selection of these majors was that the gender groups were more or less equal. Such majors as dietetics and management are more often chosen by females, and physical education or computer science by males.

Individuals in the NRF group showing concurrent education (or past education) in a health-related field were excluded from the study. Individuals who had applied knowledge and skills in rational nutrition and physical activity in their professional work were treated similarly. The physiological state of the respondent was also taken into account. Persons suffering from diseases that influence the diet and/or physical activity of the respondent (e.g., allergies, food intolerances, metabolic diseases, tumors, etc.) were excluded from the research. The same was applied to subjects who represented a specific dietary model (elimination diet or pregnancy and puerperium).

The study was limited to students in their final year of study because, in the authors’ opinion, they are highly likely to have a broad knowledge of health sciences and physical culture sciences (in the case of health students). In the case of the second group, it was also decided to include students in their final year of study so as not to disrupt the inclusion criteria and to deal with a relatively homogeneous group of students.

The study was approved by the Bioethics Committee of the Medical University of Silesia in Katowice, in light of the Act on Medical and Dental Professions of 5 December 1996, which includes a definition of medical experimentation. The study participants consciously agreed to participate in the study.

### 2.4. Research Tools

Body mass index was calculated using the formula: BMI (kg/m^2^) = body weight (kg)/height (m)^2^. The results were then interpreted using a scale [14]: ≥30.00 kg/m^2^, obesity; 25.00–29.99 kg/m^2^, overweight; 18.50–24.99 kg/m^2^, normal body weight; 17.00–18.49 kg/m^2^, underweight; and ≤16.99 kg/m^2^, malnutrition.

In the assessment of dietary intake, the author’s tool based on nutrition standards for the Polish population [15] was used, which included 20 dietary indices (e.g., frequency of consumption of individual product groups, number of meals during the day, regularity of meals during the day, snacking, fluids consumed). Respondents chose ‘yes’ or ‘no’ next to a given question about nutrition. One point was awarded for each correct answer (by the applied standards), so the highest possible total score was 20. To prioritize the results, the following scale was adopted: 18–20 points, very good nutrition; 14–17 points, good; 10–13 points, moderate; ≤9 points, poor nutrition. The questionnaire has been used previously by the authors as part of another study [16]. The tool was validated by initially sharing it with a group of 10 topic specialists. These individuals had the opportunity to add suggestions and revisions to the questions. The questions were revised according to the most common suggestions made by the specialists. The questionnaire was then made available twice to a group of 30 adults (two weeks apart). Based on the measurements, π Scott’s coefficient was calculated. For questions 1–3, 5–10, 12–15, and 18–20 a relevance of 0.93 (very good) was obtained. For questions 4, 11, 16, and 17 a relevance of 0.72 (good) was obtained.

Based on the physical activity score in the questionnaire, respondents were assigned a physical activity index (PAL) based on current recommendations for physical activity [14]: 1.2, no physical activity; 1.4, low physical activity (approximately 140 min per week); 1.6, medium physical activity (approximately 280 min per week); 1.8, high physical activity (approximately 420 min per week); and 2.0, very high physical activity (approximately 560 min per week).

The PSS-10 is used to assess the intensity of stress related to one’s living situation over the past month. The scale is designed mainly for research purposes and can be used in practice, screening, prevention, and assessing the effectiveness of therapeutic interventions. Scores from 0 to 13 are considered low, while scores of 20 and above are considered high. Internal consistency was checked in a study of a 120-person group of adults, yielding a Cronbach’s alpha index of 0.86. The correlation of all questions with the overall scale score is satisfactory. Reliability determined by testing a group of 30 students twice at an interval of two days was 0.90, and at an interval of four weeks was 0.72 [17].

With the TFEQ-13, it is possible to assess three behaviors using 13 questions comprising three subscales: five questions relate to eating restriction (questions 1, 9, 10, 12, and 13), five questions relate to lack of control over eating (questions 2, 5–7, and 11), and three questions are directly related to eating under the influence of emotions (EE) (questions 2, 4, and 8). The questionnaire contains standardized answers on a four-point scale ranging from zero to three. The respondent marks the most defining statement next to each sentence: ‘definitely yes’, ‘rather yes’, ‘rather no’, and ‘definitely no’. Values are calculated separately for each subscale. The higher the score obtained, the higher the strength of the behavior. Internal consistency alpha Cronbach’s coefficient for the entire scale was 0.78, and for the subscales it was 0.78 for eating restriction, 0.76 for lack of control over eating, and 0.72 for eating under the influence of emotions. All subscales correlated with each other significantly positively (*p* < 0.001) [18], of which only the score indicating EE was used in the present study.

### 2.5. Study Procedure

The study consisted of a survey questionnaire and an album of sample foods and dishes. The study was conducted according to scientific ethics, anonymity rules, and the RODO clause (Polish Law on Respect for Classified Information). The survey was conducted using an online form, which is an acceptable method in psychological research. The link to the questionnaire was distributed to participants using email boxes dedicated by the university. During data collection, methods were used to prevent fake/bot responder phenomenon by checking login times and questionnaire completion times. In addition, the questionnaire was secured with a CAPCHTA key. The questionnaire of the survey consisted of a metric (data of the subject: gender, age, a field of study and occupation, and anthropometric data—declared height and body weight); the author’s questionnaire of dietary habits based on the guidelines and standards of the National Institute of Public Health and the National Center for Nutrition Education [15]; questions about physical activity practiced and its level based on WHO guidelines [14]; the Perceived Stress Scale—PSS-10 (polish adaptation) [19]; and the Three-Factor Food Questionnaire (TFEQ-13). The survey questionnaire was available online May–June 2021.

In the second stage of the study, respondents were presented with a scrapbook containing sample foods and dishes. An album of sample foods and dishes was used to verify the ability to estimate the size and calorie content of portions, consisting of 12 photographs consistent with the division of foods into 12 groups (one photograph per group) [18]. The study using the album was conducted with the sensory panel of the Department of Dietetics, Faculty of Health Sciences in Bytom, Silesian Medical University in Katowice, Poland, July–August 2021. Before each study, visual perception (perception of images) was tested using a scrapbook. For this purpose, selected Ishihara boards and optical illusion boards were used. Both tools are commonly used to assess so-called visual daltonism and the perception of objects in pictures (e.g., size, shape, length). To link the results of the questionnaire with the album, each participant of the study was given an individual number while filling in the questionnaire, which was then also entered into the album.

### 2.6. Statistical Analysis

Tables were prepared for all extracted data from the survey questionnaire, and descriptive statistics (percentages (%), counts (N; n), mean (X), standard deviations (SD)) were calculated. Detailed statistical analyses were conducted, regarding the demonstration of differences between the represented behaviors (pro-health or anti-health) and the occurrence of EE in the sample group. To analyze the above material, the chi-square (χ^2^) test and the V-Cramér (V) coefficient of the strength of the relationship (with Yates and Fisher’s correction) were used. A probability level of *p* = 0.05 was assumed for the study.

## 3. Results

The gender of the subjects was as follows: women, 60.0%, 184 subjects (HRF: 30.6%, n = 92; NRF: 30.6%, n = 92); men, 40.0%, 116 subjects (HRF: 20.6%, n = 62; NRF: 18.2%, n = 54). The age of the subjects was 26 years (±2 years). More than 269 persons (89.9%) lived in large cities (defined as more than 100,000 residents), 23 persons (7.5%) lived in smaller cities (defined as less than 100,000 residents), and 8 persons (2.6%) lived in rural areas. Only 13.1% of respondents (38 persons) had permanent employment, i.e., in the telecommunications, service, and administrative/office sectors. Of the surveyed group, 75.6% had an income of an average level, 12.2% had an above-average income, and 12.2% had a below-average income (the minimum wage in Poland in 2021 was PLN 3010—about €630). Statistically, the groups did not differ in the above variables (Table 1).

Regarding BMI, more than 15.2% of the subjects were characterized as underweight (44 subjects in the HRF group). Normal weight was a characteristic of 178 subjects (61.3%). Overweight and obesity were present only in the NRF group, with a total of 68 subjects (23.4%). Based on the results of the dietary assessment, it was found that the best dietary model was characterized by the HRF group; in this group, 97.2% of students were characterized by a very good and good dietary mode (84.0%, 121 persons; 13.4%, 19 persons, respectively). The NRF group, on the other hand, was dominated by sufficient dietary mode, at 64.4% of all cases in this group (94 persons). Less popular was the dietary model marked as “good”, with only 24.6% of this group (36 persons). It should be emphasized that an incorrect dietary pattern was represented only by persons from the HRF group (3.9% of the total number of subjects, 11 persons).

Low physical activity in the PAL index was characteristic for 46.2% of respondents (122 persons), and most often chosen by persons from the NRF group (79.5%, 97 persons). Medium physical activity was observed in 25.7% of the respondents (68 persons); this activity concerned both the HRF group (33.8%, 48 persons) and the NRF group (16.4%, 20 persons). Physical activity at a high and very high level concerned 28.1% of the students (75 persons). These were mainly persons from the HRF group (48.4%, 70 persons). However, two individuals from the HRF and 24 individuals from the NRF group did not engage in any physical exercise daily (1.4% vs. 16.4%).

Taking into account the test of estimating the size and caloric content of portions, it was found that 32.4% (94 persons) overestimated the size of the portions of products and dishes indicated in the photographs. In this group, there were mostly persons studying in health faculties (57.6%, 83 persons); less often, there were persons from other faculties (7.5%, 11 persons). In the case of underestimation (33.8%, n = 98), the situation was reversed—persons from the NRF group mainly underestimated the size of products and dishes (56.2%, n = 82); in the HRF group, much fewer persons underestimated (11.1%, n = 16). The remaining persons correctly indicated the size of the portion (33.8%, 98 persons). Analyzing the results of the test on the ability to estimate the calorie content of portions based on photographs, it was observed that 35.8% (104 persons) overestimated the calorie content of the products and dishes indicated in the photographs. This group included mainly health-related persons (58.3%, 84 persons), and less frequently, non-health-related persons (13.0%, 19 persons). On the other hand, in the case of underestimation of the energy of dishes (35.2%, n = 102), persons from the NRF group mostly underestimated the caloric value of products and dishes presented in the album (55.5%, n = 81); in the HRF group, such cases were much less (15.3%, n = 22). The remaining persons correctly indicated the calorie content of the portion (29.0%, n = 84).

The respondents’ level of perceived stress was measured twice—before and after the survey—and since no statistically significant relationship was shown between the measurements, it was decided to average these results (*p* > 0.05). Analysis of the PSS-10 questionnaire showed that 86.7% (130 persons) of the HRF group and 46.7% (70 persons) of the NRF group had low levels of stress. Correspondingly, 23.3% (20 persons) in the HRF and 53.3% (80 persons) in the NRF show elevated levels of stress. One question of the scale concerned the frequency of stressful situations that exceed the body’s resilience and result in feelings of discomfort, aggression, and jitteriness. Both the HRF and NRF groups had an average score measuring between two and three points—2.41 for the HRF and 2.56 for the NRF, which indicates that in the frequency of stressful situations in life the group can be considered homogeneous.

Based on the results of the TFEQ-13, among 120 subjects (41.4%) there were behaviors consistent with limiting food intake (HRF, 72.4%; NRF, 11.0%), while 64 subjects (20.7%) were characterized by a lack of control over food intake (HRF: 13.8%, 20 subjects; NRF: 27.4%, 20 subjects). Emotional eating was characteristic of 106 students (37.9%), with the NRF group dominating (61.6%, n = 90). It was observed that a high PSS-10 score is mainly characteristic of individuals who exhibit EE (χ^2^ = 10.279; V = 0.731; *p* = 0.001): PSS-10 average score in HRF group was 29 ± 2 and NRF group it was 34 ± 2 (χ^2^ = 11.893; V = 0.657; *p* = 0.001). Slightly higher high-stress scores were observed in representatives of the NRF group. These results were compared with those of PSS-10, and details of the analysis are shown in Table 2.

Another analysis concerns the group in which EE behavior was demonstrated (n = 106). High BMI values were present in the NRF group, indicating a statistically significant relationship between the indicated characteristics (χ^2^ = 13.238; V = 0.723; *p* = 0.0001). Similarly, the same was true for diet. Individuals representing a good (27.6%) and very good (16.2%) diet are less likely to belong to the group of those with an increased risk of emotional eating. In this case, there was also a statistically significant correlation associated with NRF group membership (χ^2^ = 10.984; V = 0.683; *p* = 0.0001). Next, it was decided to verify the relationship between the occurrence of emotive eating and the level of physical activity represented. Based on the statistical inference performed, it was found that low PAL values were present in NRF subjects, indicating the presence of a statistically significant relationship between the indicated characteristics (χ^2^ = 8.117; V = 0.597; *p* = 0.002). On the statistical analyses conducted, it should be concluded that both in the case of estimating portion size and caloricity of meals there is a statistical relationship: NRF subjects characterized by emotional eating are more likely to underestimate the size and caloricity of the meal (χ^2^ = 12.467; V = 0.601; *p* = 0.0001/χ^2^ = 11.551; V = 0.582; *p* = 0.0001). The last verification concerned the relationship between the occurrence of emotional eating in the study group and the level of perceived stress. Higher levels of stress have been shown to occur in NRF individuals (χ^2^ = 9.963; V = 0.699; *p* = 0.015)—Table 3.

## 4. Discussion

Under conditions of prolonged negative emotions, vulnerable people take action to change a given situation and improve their mental state. Often these actions are not taken consciously but are only performed intuitively. One way to intuitively cope with stress is to reach for food, as hunger is often mistaken for feelings of stress. Stress is closely linked to nutrition, not least because the elevated cortisol levels in this state trigger feelings of hunger [20]. In addition, stress increases the demand for serotonin (5-hydroxytryptamine), which in turn results in an increased demand for carbohydrates, which affect the release of endorphins and increase serotonin synthesis [21].

According to Kosicka-Gębska et al., 22% of the Polish population reaches for sweets in situations that cause stress [22]. The reason that sweets are the most common choice of food during stress is related to several factors. One of them is the increased need for carbohydrates [21,23,24]. Stress accelerates the breakdown of serotonin, so it is more common to feel the urge to introduce sugars into the body to make up for serotonin deficiencies. However, the soothing effect of serotonin is temporary, lasting about three hours, and once its levels are reduced again, the desire for sweet foods is restored [25]. Carbohydrates activate insulin, which in turn stimulates the brain to produce tryptophan, a precursor to serotonin. When the body’s serotonin levels drop, people may feel depressed or stressed. Therefore, they reach for sugary foods, which will again cause serotonin to be secreted and reduce feelings of stress [26]. The effect of serotonin is that people feel calmer and sleepier; people stop thinking about stress after a meal rich in carbohydrates and scant protein [27].

The author of the term ‘emotional eating’, Hilda Bruch [28,29], as well as many modern researchers [30,31,32], assume that excessive food intake under the influence of emotions is the result of a failure to adequately distinguish between physiological hunger signals and emotional hunger. This results in excessive food intake (to reduce negative emotions) and weight gain [33,34]. Another explanation is the vicious cycle mechanism of food-mediated emotion regulation, according to which negative emotions are the source of physiological stimulation misidentified as feelings of hunger. This stimulus contributes to the immediate consumption of food, which consequently leads to a temporary reduction in negative emotions. Subsequently, the level of negative emotions increases again, which is associated with further food intake and progressive weight gain [35,36].

Van Strien et al. studied the relationship between emotions (joy and sadness) and eating, with highly emotional participants eating significantly more in a sad mood than in a joyful mood [37]. Macht et al. found that before the exam, participants in the stress group reported higher ratings of negative emotions (tension, fear, or emotional distress) and lower ratings of positive emotions (happiness, relaxation, or positive mood), with a corresponding higher tendency to eat to distract from the stress. These findings underscore the importance of capturing positive and negative emotions when examining associations with eating behavior [38]. Not only can positive emotions be associated with eating, but also how the absence of pleasant emotions when experiencing unpleasant emotions can have an impact [37]. The finding that positive emotions can influence eating makes intuitive sense when considering the use of food as part of social rituals (such as birthdays, weddings, and religious events) [39]. The implication is that while individuals do not necessarily use food to regulate positive emotions, positive emotions can trigger increased consumption through associative learning. Alternatively, a positive emotional state can divert attention from the source of positive emotions, interfering with the conscious reduction of food intake [40].

Stress is the most commonly studied emotion in assessing eating behavior. Both chronic stress and temporary stress have been associated with higher food intake, with people eating more during periods of stress [41,42,43,44]. However, depression and sadness have also been reported as antecedents of eating behavior [45]. Boredom and emotional eating showed strong positive correlations [46], and people who were ashamed ate more in a taste test experiment, with no effect on guilt [47]. Aggression and anger were positively correlated with emotional eating [48].

Emotions were also found to influence the type of food eaten. Feelings of stress influenced food choices toward more palatable and less healthy meals [49,50], while motivation to eat to regulate negativity was associated with an unhealthy eating pattern [51]. These findings suggest that negative emotions can trigger unhealthy eating behaviors, and poor food choices, and food is used to regulate stressful or negative emotions in healthy and overweight individuals. If we consider the demands of daily life, using unhealthy foods to regulate negative emotional states can contribute to a steady increase in weight over time.

Individual differences also appear to be highly significant. For example, people with higher dietary restriction scores feel a greater desire to eat when asked to accept or suppress their emotions [52]. Sleep quality also requires attention. The combination of sleep deprivation and a propensity for emotional eating has been associated with increased food consumption [53]. Short sleep is thought to increase hunger and appetite through the effects it has on leptin and ghrelin, and sleep deprivation itself may act as an additional stressor [54,55]. Thus, poor sleep can potentially undermine or cancel out the effects of any intervention. Conversely, it is possible that adopting a good sleep pattern is part of an intervention that may enhance the effect of other interventions [56].

The study by Bennett et al. determined the relationship between emotional eating behaviors and the tendency to eat palatable foods among college students aged 19.6 ± 1.0 years. The mean BMI of the subjects was 24.1 ± 1.2 kg/m^2^. There was a positive correlation between BMI and negative emotions and negative situations (*p* < 0.01). A one-unit increase in BMI resulted in a 0.293-unit increase in negative situation scores and a 0.626-unit increase in negative emotions scores [57].

Greene et al. found that college students who scored high on emotional eating had higher BMIs than students who scored lower on emotional eating. Therefore, understanding the importance of emotional eating may be particularly important in preventing weight gain during college, which can lead to obesity in adulthood [58]. Students frequently cited happiness and stress as the two most frequently experienced emotions. The main source of stress cited by students was school [59]. In particular, studying for exams, completing school assignments, and time management were cited as sources of stress.

Stress affected the eating behavior of men and women, but in opposite ways. Women increased consumption when they were stressed about school—as many as 62% of the women surveyed gave this answer. Under normal conditions, 80% reported making healthy food choices, but in a stressful situation, only 33% ate healthily [60]. A study conducted on a group of Australian students by Papier et al. found that more than half (52.9%) of the students suffered from stress, with relatively more women (57.4%) than men (47.4%). Female students who experienced mild to moderate stress were 2.22 times less likely to eat processed foods (*p* < 0.01) than non-stressed female students. Men who experienced mild to moderate levels of stress tended to eat more highly processed foods (*p* < 0.05) and drink more alcohol (*p* < 0.05) than non-stressed male students [61]. This may suggest a decrease in healthy food intake and an increase in unhealthy food intake during periods of emotional stress-eating.

A study by Lazarevich et al. examined the relationship between depressive symptoms, emotional eating, and BMI in Mexican college students. They found that depressive symptoms were associated with emotional eating in both men (*p* < 0.001) and women (*p* < 0.001), while emotional eating was associated with BMI and men (*p* < 0.001) and women (*p* < 0.001). The indirect effect of depression through emotional eating on BMI accounted for a significant portion of the total effect in both men (23.1%) and women (25%) [62].

Students are a high-risk group for the development of emotional eating disorders due to their exposure to numerous factors, i.e., stressful situations, peer pressure, and lack of time for physical activity. Fields of study related to ‘health’ in the broadest sense shape behavioral patterns that influence the maintenance of mental as well as physical health. Therefore, the results obtained in our study indicate better indicators in the aspect of emotional nutrition in this group.

## 5. Strengths and Limitations

The research on the prevalence of emotional eating among students of different majors allowed us to understand the basic mechanisms, cause–effect relationships, and determinants of the occurrence of the indicated disorders. Conducting the research required much work and preparation in the form of developing research tools and becoming familiar with existing psychometric tools measuring the risk of emotional eating. Of course, the paper does not suggest that it is the field of study that influences the development of the disorder, but rather that individuals who choose it are characterized by certain traits that predispose them to it. This should be understood in the way that, thanks to the results of the study, it is possible to detect groups of persons who should be included in the observation in terms of the control and safety of their lifestyle. An important limitation of the study is that causation cannot be described since it is a cross-sectional study. The main difficulties during the conduct of the study were access to the student group because the study was conducted in the period of May–June 2021, and it should be emphasized that this was the period immediately adjacent to the lifting of the COVID-19 pandemic restrictions.

An important part of the next research should be to answer the questions: how should study participants deal with stress, and what coping mechanisms should they use to avoid dealing with stress and negative emotions?

## 6. Conclusions

The results obtained in the study indicate that there is a relationship between the respondents’ lifestyles and the occurrence of emotional eating. Individuals who have elevated BMI values, unhealthy diets, low physical activity rates, and underestimate the size of their meals in terms of weight and calories, as well as experiencing high levels of stress, are more likely to develop emotional eating. These results also indicate that further research in this area should be undertaken to indicate whether the relationships shown can be generalized.

## Figures and Tables

**Table 1 nutrients-14-03289-t001:** Comparison of the studied groups (N = 300; HRF = 150; NRF = 150).

Group		HRF	NRF	Total	χ^2^	*p*-Value
**Gender**	**Female**	92 (30.6%)	92 (30.6%)	184 (61.2%)	21.391	*p* > **0.05**
	**Male**	62 (20.6%)	54 (18.2%)	116 (38.8%)	29.122	
**Age**		26 ± 2 *	26 ± 2 *	26 ± 2 *	18.974	
**Residence**	**Large city**	139 (46.3%)	130 (43.6%)	269 (89.9%)	32.004	
	**Small city**	12 (4.0%	11 (3.5%)	23 (7.5%)	35.680	
	**Rural area**	3 (1.0%)	5 (1.6%)	8 (2.6%)	31.404	
**Income**		PLN 3000 ± 500 * (€600 ± 120) *	PLN 3000 ± 450 * (€600 ± 100) *	PLN 3000 ± 475 * (€600 ± 110) *	28.901	

HRF, health-related field; NRF, non-health-related field; χ^2^, chi-square test; * mean ± standard deviation.

**Table 2 nutrients-14-03289-t002:** Comparison of PSS-10 and TFEQ-13 scores in the study group (N = 300; HRF = 150; NRF = 150).

Group			HRF		NRF		Total	χ^2^	V	*p*-Value
			All	Only EE Cases (by TFEQ-13)	All	Only EE Cases (by TFEQ-13)				
**PSS-10**	**Low** **perceived stress**	200 (66.67%)	130 (86.70%)	8 (8.48%)	70 (46.70%)	19 (20.14%)	200 (66.67%)	12.113	0.611	**0.003** *
	**Average score**		7 ± 1	9 ± 2	11 ± 1	12 ± 1	10.8 ± 0.9	11.244	0.522	**0.002** *
	**High** **perceived stress**	100 (33.33%)	20 (23.30%)	8 (8.48%)	80 (53.30%)	71 (75.26%)	100 (33.33%)	10.279	0.731	**0.001** *
	**Average score**		27 ± 1	29 ± 2	32 ± 2	34 ± 2	30.8 ± 1.0	11.893	0.657	**0.001** *

HRF, health-related field; NRF, non-health-related field; χ^2^, chi-square test; V, V-Cramer; PSS-10, perceived stress scale; EE, emotional eating; TFEQ-13, Three-Factor Eating Questionnaire; * *p*-value statistical significance.

**Table 3 nutrients-14-03289-t003:** Scores of subscales on emotional eating by selected indicators (N = 106; HRF = 16; NRF = 90).

Group		HRF	NRF	Total	χ^2^	V	*p*-Value
**BMI**	**Malnutrition**	0	0	0	13.238	0.723	**0.0001** *
	**Underweight**	0	0	0			
	**Normoweight**	10 (10.60%)	22 (23.32%)	38 (40.28%)			
	**Overweight**	6 (6.36%)	55 (58.30%)	61 (64.66%)			
	**Obesity**	0	7 (7.42%)	7 (7.42%)			
**Diet quality**	**Poor**	0	3 (3.18%)	3 (3.18%)	10.984	0.683	**0.0001** *
	**Moderate**	2 (2.12%)	72 (76.32%)	74 (78.44%)			
	**Good**	14 (14.84%)	15 (15.90%)	29 (30.74%)			
	**Very Good**	0	0	0			
**PAL**	**PAL 1.4**	4 (4.24%)	34 (36.04%)	3 (3.18%)	8.117	0.597	**0.002** *
	**PAL 1.6**	12 (12.72%)	31 (32.86%)	43 (45.58%)			
	**PAL 1.8**	0	25 (26.50%)	25 (26.50%)			
	**PAL 2.0**	0	0	0			
**Portion size**	**Underestimation**	0	56 (59.36%)	56 (59.36%)	12.467	0.601	**0.0001** *
	**Estimate**	16 (16.96%)	34 (36.04%)	50 (53.00%)			
	**Revaluation**	0	0	0			
**Caloric size**	**Underestimation**	6 (6.36%)	45 (47.70%)	51 (54.06%)	11.551	0.582	**0.0001** *
	**Estimate**	10 (10.60%)	34 (36.04%)	44 (46.64%)			
	**Revaluation**	0	11 (11.66%)	11 (11.66%)			
**PSS-10**	**Low**	8 (8.48%)	19 (20.14%)	37 (39.22%)	9.963	0.699	**0.015** *
	**High**	8 (8.48%)	71 (75.26%)	79 (83.74%)			

HRF, health-related field; NRF, non-health-related field; χ^2^, chi-square test; V, V-Cramer; BMI, Body Mass Index; PAL, physical activity level; PSS-10, perceived stress scale; * *p*-value statistical significance.

## Data Availability

The original contributions presented in the study are included in the article/supplementary material; further inquiries can be directed to the corresponding author.

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
