# Peer review of "Prevalence of Emotional Eating in Groups of Students with Varied Diets and Physical Activity in Poland"

_nutrients, 2022, doi:10.3390/nu14163289_

Round 1

Reviewer 1 Report

My comments/questions to the authors are as follows:

1.      Why were the subjects restricted to only final -year students?

2.      Did the questionnaire involve socio economic background of the students?

3.      Were the stress levels of the participants measured before getting involved in the study? like how often they have been upset about something or felt like they were unable to control things in life?

4.      How did the participants cope up with the stress? Any coping mechanism such as meditation, yoga which helped their eating behavior?

Author Response

Dear Reviewer,
Thank you for conducting a review of the manuscript. We are grateful for your valuable advice and thorough suggestions. We have addressed all issues, the required corrections have been made in the manuscript and marked in red. In addition, a reference in the form of a commentary has been inserted next to each amendment to make it easier to read the changes made.

Once again, we sincerely thank you,
Authors

Reviewer 2 Report

Dear Authors,

To enhance the readability and wider context for the audience, I have some points below which I believe should be addressed to make the paper stronger.

Abstract:

I strongly propose to delete the sentence "The main exposure group ... (lines 25-27) as this is not the purpose of the study.

This section should include the type of study (cross-sectional? Other?)

The age and % of men and female should be reported.

Introduction:

Lines 83-85: Literature source should be cited.

Does gender (or other factors, e.g. socio-economic status, education etc.) matter in the problem of emotional eating? Some more information on factor influencing EE should added.

If emotional eating is not a specific eating disorder, are there appropriate diagnostic options? Authors should add information in this regard.

Materials and methods:

I appreciate that the authors referred to the previous article, but for the convenience of the reader it is worth including the study scheme (e.g. CONSORT chart) in the manuscript or additional material.

On what basis was the number of participants determined? Information on this is missing?

The abbreviation should be explained: ZCEZ (line 169).

How were respondents recruited? Who was the information / request to participate in the study addressed to? What was the percentage of refusal? All this information is missing and it may be important to assess whether the group was representative (this information is also missing).

Line 191:  It should be “ 30 kg/m2..” 

Incorrect BMI criteria - normal body weight is between 18.5 and 24.99; underweight is < 18.5. The authors need to correct this as it affected the results and the inference that might be wrong!

Could Authors explained more on the tool used to assess dietary intake of respondents as the reference quoted is Polish Nutritional Norms. They contain recommended amounts of energy and nutrients for different population groups, and the authors did not assess the amount of energy and nutrients consumed?  

Results:

Tables should be better described: the reader must seek explanations of the abbreviations in the text in order to understand the results.

The authors did not explain how they assessed the respondents' energy consumption. Table 2 shows the number of respondents who correctly or incorrectly assessed their energy consumption - the reader does not know on what basis it was assessed?

Discussion:

It is difficult for me to relate to the discussion if I believe that the misinterpretation of the BMI values had a large impact on the results of the study. The authors should re-analyze the results and conduct a discussion based on them.

Conclusions:

On the basis of the conducted research, the authors cannot state whether lifestyle can have a real impact on the development of emotional eating problems. At most they can conclude that there is a relationship between these factors.

Numerous irregularities in the bibliography!

Author Response

Dear Reviewer,
Thank you for conducting a review of the manuscript. We are grateful for your valuable advice and thorough suggestions. We have addressed all issues, the required corrections have been made in the manuscript and marked in blue. In addition, a reference in the form of a commentary has been inserted next to each amendment to make it easier to read the changes made.

 In addition, linguistic and editorial corrections have been made, the bibliography has been corrected and all the imperfections of the text pointed out in the review have been addressed point by point.

Once again, we sincerely thank you,
Authors

Round 2

Reviewer 2 Report

Dear Authors,

Thank to Authors for some of my comments. 

However, I still belive that:

- if Authors used own questionaire to assess the correctness of the diet, information should be provide on this questionaire (e.g. validation?)

- I wonder how it is possible that a major change in BMI criteria (in previous version BMI criteria for normal body weight 20-24.99 and in present version 18.5-24.99) did not affect the number / % of individuals with normal body weight (178 individuals/ 61.3% in both versions) and underweight? 

All results concerning the number of individuals with different body weight status are the same in previous and present version of the manuscript with different criteria of BMI (table 2)?

Author Response

Dear Reviewer,

The results of the study were not altered due to the fact that the BMI ranges were selected correctly from the outset, as the values were calculated in a diet calculator that incorporates the current WHO guidelines relating to BMI. What occurred was merely an editorial error when describing the methodology. At this point, we thank you very much for catching this error, which escaped the authors in the flurry of work. We had already explained it during the first review, but only now did we notice that the MDPI system does not pass Word files containing comments (as an aside). As proof, we attach a PDF with visible comments (from the previous round). We are very sorry for the situation.

Regarding the questionnaire issue, we have added information on validation and prior use of the tool to the newly uploaded version.

Greetings.
